# PHYSICS OF LANGUAGE MODELS: PART 2.2, HOW TO LEARN FROM MISTAKES ON GRADE-SCHOOL MATH PROBLEMS

[EXTENDED ABSTRACT]*

**Tian Ye**
FAIR at Meta and Carnegie Mellon University
`sigma648@meta.com` and `tye2@andrew.cmu.edu`

**Zicheng Xu**
FAIR at Meta (now Google Research)
`zichengxu@meta.com` (now `zichengxu@google.com`)

**Yuanzhi Li**
Mohamed bin Zayed University of AI
`Yuanzhi.Li@mbzuai.ac.ae`

**Zeyuan Allen-Zhu**
FAIR at Meta
`zeyuanallenzhu@meta.com`

## ABSTRACT

Language models have demonstrated remarkable performance in solving reasoning tasks; however, even the strongest models still occasionally make reasoning mistakes. Recently, there has been active research aimed at improving reasoning accuracy, particularly by using pretrained language models to "self-correct" their mistakes via multi-round prompting. In this paper, we follow this line of work but focus on understanding the usefulness of incorporating "error-correction" data directly into the pretraining stage. This data consists of erroneous solution steps immediately followed by their corrections. Using a synthetic math dataset, we show promising results: this type of pretrain data can help language models achieve higher reasoning accuracy directly (i.e., through simple auto-regression, without multi-round prompting) compared to pretraining on the same amount of error-free data. We also delve into many details, such as (1) how this approach differs from beam search, (2) how such data can be prepared, (3) whether masking is needed on the erroneous tokens, (4) the amount of error required, (5) whether such data can be deferred to the fine-tuning stage, and many others.

## 1 INTRODUCTION

Language models have achieved near-human-level performance in various tasks, including math solving, coding, and natural language understanding (Allen-Zhu & Li, 2023; Olsson et al., 2022; Ahn et al., 2024; Zhou et al., 2023; Raghu et al., 2021), and have been argued to have reached an L2 or L3-level of intelligence (Allen-Zhu & Xu, 2025). However, their problem-solving skills are still imperfect, sometimes resulting in logical errors. Recently, there have been numerous attempts to improve the reasoning accuracy of language models.

---

*The first six papers in the *Physics of Language Models* series were presented as a two-hour tutorial at ICML 2024 in Austria (`youtu.be/yBL7J0kgldU`). A 50-min deep dive into Part 2.2 is available at `youtu.be/yBgxxvQ76_E`. Full and future editions of Part 2.2, including code release, are available at `physics.allen-zhu.com`, `github.com`, and `ssrn.com/abstract=5250631`.

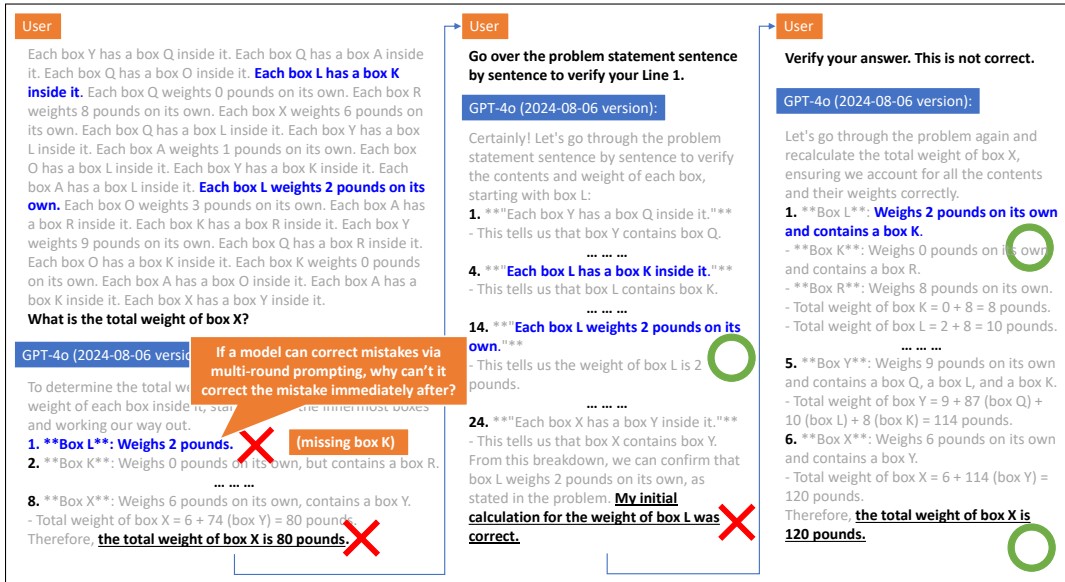

Figure 1: An illustration of how GPT-4o self-verifies and corrects its own mistakes.
**Observation:** Correcting mistakes during generation, rather than after, can save inference tokens (the model avoids generating useless tokens following an erroneous step) and simplify the inference process (eliminating the need for multiple rounds of verification and correction).

One promising approach is to use a verifier to check the correctness of the language model's output (Li et al., 2023a; Cobbe et al., 2021; Zhang et al., 2024; Skreta et al., 2023; Yang et al., 2022). Interestingly, some studies show that language models can "self-verify" (Weng et al., 2022; Madaan et al., 2024): They can be prompted to verify the correctness of their own generation, thereby improving overall accuracy. An illustrative example is shown in Figure 1.

This leads to the following fundamental questions:

*If a language model can correct its own mistakes after generation, (1) Why does it make those mistakes to begin with? (2) Why doesn't it correct the mistakes immediately during generation, instead of waiting until after?*

There are many works that attempt to understand question (1). Notably, studies such as (Shu & Yu, 2024; Miller et al., 2020; Anil et al., 2022) show that many mistakes arise due to "distribution shift" — the training distribution of the language model differs from the prompts used during testing. Thus, even if the training data is error-free, language models can still make mistakes during generation.

Much less work focuses on question (2). While correcting mistakes after generation is a valid approach to improve a language model's accuracy, it is more desirable to correct mistakes immediately as they occur, **such as "$A \Rightarrow B$, oh I made a mistake, actually $A \Rightarrow C$."** Doing so during generation can save inference tokens (the model does not need to continue generating based on an erroneous step) and simplify the inference framework (eliminating the need to call the model multiple times to verify and correct the answer), see Figure 1. Thus, can we train language models to directly perform (2)?

A natural conjecture for why current language models cannot perform (2) is an alignment issue: There might be little to no training data in the language model's corpus to teach it to correct errors immediately. After all, people are unlikely to write an erroneous sentence followed by immediate correction in internet articles.[1] However, **even if** we are given enough such "**retry**" training data (i.e., error + immediate correction such as "$A \Rightarrow B$, oh I made a mistake, actually $A \Rightarrow C$"), **is it clear that the language model can improve its overall reasoning correctness?**

- **Potential harm of next-token prediction on data with mistakes:** Training on data with mistakes can be dangerous. By minimizing the next-token prediction objective, the model might learn to generate mistakes (such as "$A \Rightarrow B$") instead of writing the correct steps (i.e., $A \Rightarrow C$).

---

[1]If they make an error when writing an article and realize it afterward, they will simply delete the error line in the final article.

One could try to mitigate this by masking out the erroneous steps. However, unlike the "error correction after entire generation" type of training data where we can simply mask out the entire generation,[2] here, the error and correction steps are intertwined, making it much harder to mask out the error only.

- **Unclear benefits over training on "perfectly error-free" data:** Intuitively, if our final goal is to have model output correct reasoning steps to improve the reasoning accuracy, then why would training on "$A \Rightarrow B$, oh I made a mistake, actually $A \Rightarrow C$" be better than training directly on the correct step "$A \Rightarrow C$"? Introducing errors is a distribution shift compared to what we want the model to generate during test time (i.e. a solution without error). Moreover, if we mask out the error part [$A \Rightarrow B$, oh I made a mistake, actually], would it just reduce to training on perfectly correct data, with the model simply learning $A \Rightarrow C$?

In this paper, we provide initial results towards understanding the usefulness of including training data in the language model's corpus that teaches it to correct errors immediately. Specifically, we ask the following question:

> Can training on retry data (errors and immediate corrections) successfully teach the language model to perform error correction? **Can a language model achieve higher reasoning accuracy compared to training on the same amount of perfectly error-free data**?

To initiate our study and perform a controlled experiment, we need a setting where we can reliably collect data with errors and corrections. While it is tempting to use language models such as GPT-4 to synthesize such data, there is no evidence that GPT-4 can robustly generate errors or make those corrections consistently.[3] To avoid complicating our conclusions with the success rate and reliability of GPT-4, we need a different setting where we can 100% reliably generate errors and corrections.

In this work, we choose to use the iGSM dataset (Ye et al., 2025), which consists of a large number of program-generated (not LLM-generated) elementary-school level math reasoning problems. We discuss the dataset and provide examples in Section 2 to make this paper self-contained. Due to the generation procedure, we can also easily construct erroneous steps (simulating the reasoning mistakes that GPT-4 can make on such problems) and make 100% correct corrections to those steps. We choose this setting because mathematical reasoning errors are among the most widely observed errors made by language models. We outline our results in the following section.

**Results 0-1: retry upon regret.** We begin with a warmup result. After pretraining the model on perfectly *error-free* math data, one can finetune it with a small number of trainable parameters (e.g., a rank-8 update on its embedding layer) to detect errors in its solution steps. This finetuning process is remarkably lightweight and essentially indicates that the model "already knows" it has made a mistake (i.e., its internal states exhibit regretful behavior).[4]

Next, we consider a "retry upon regret" generation process. During the inference phase, if the model detects an error in its generated solution step, it will "regenerate" from the end of its previous sentence. We demonstrate that this process leads to improved accuracy on this math dataset, surpassing what can be achieved using beam search.

*Remark* 1.1. This should not be confused with the "self-correction" works (Pan et al., 2023; Huang et al., 2023; Madaan et al., 2024), where the model evaluates its own output through careful prompting. Here, we finetune the pretrained model for error detection ($> 99\%$ accuracy), and this error detection accuracy is significantly higher than that achieved through prompting.

---

[2] For example, we can have training data like [MaskStart] A wrong math solution [MaskEnd] What is the error in the above solution? ...

[3] Although GPT-4 can sometimes "verify" the correctness of a generated solution, its verification accuracy is far from 100%. In the example of Figure 1, using prompts such as "Please verify carefully and correct your solution," "Please check your solution line by line carefully, matching it to the problem description," or even "You made a mistake in Line 1, please correct it," or "I am very sure you made a mistake in Line 1, please correct it by checking it against the problem statement carefully," GPT-4o can insist that its solution is correct and makes no correction (using the 2024-08-06 version with temperature 0).

[4] Similar observations have been made in (Liu et al., 2023; Li et al., 2023b; Wang et al., 2024), where a pretrained transformer can be easily finetuned to effectively verify its own solutions (true or false).

While this provides strong evidence that language models "regret" their mistakes, improving final accuracy relies solely on randomness (for re-generation) to correct errors — similar to beam search.[5] This can take many rounds to re-generate, which is inefficient and requires the error detector to be highly accurate. It also alters the generation process, which might not align with the "general intelligence" framework, where the goal is typically for one model to perform all tasks using the same (autoregressive) decoding algorithm.

**Results 2-6: pretrain with retry data.** We now turn our attention to *retry data*. If the pretraining data includes errors and their immediate corrections (which we call "retry data"), the model can learn to not only detect errors but also correct them, achieving much higher accuracy compared to "retry upon regret" in Section 3. This differs from the "self-correction" line of work: the model learns to *automatically retry* after detecting an error, without requiring additional prompting or multi-stage generation.

Furthermore, we discover that even when the model is pretrained entirely on retry data with high error rate (e.g., for the iGSM data, $p = 20\%$ or even $p = 50\%$, meaning roughly half of the solution steps have an inserted error), it does not tend to produce erroneous steps during generation. The model still strives to generate "perfectly correct" solutions most of the time and only corrects itself on the rare occasions when it makes a mistake, leading to an overall improvement in accuracy. In fact, within a reasonable range, the higher the $p$ value in the pretraining data, the better the model performs in reasoning accuracy.[6] We also observe that it is not necessary to perform label masking on the errors, so the vanilla autoregressive (causal masking) training simply works.

**Result 7: finetune with retry data.** In contrast, if a model is already pretrained with error-free data, even using sufficiently many (additional) retry data, with a wide range of LoRA finetuning configurations, label masking or not, the model's accuracy does not significantly improve. This indicates that the skill of error correction can be very different from the original error-free reasoning, and thus requires major weight changes, potentially beyond what parameter-efficient fine-tuning (PEFT) can handle. For comparison, we show that full finetuning is effective when sufficiently many retry data is available, though this resembles continued pretraining.

Thus, unlike error detection (see Section 3, where even a rank-8 adaptation on the embedding layer suffices), **error correction** is **not** a skill that can be easily adapted from a model pretrained with only error-free data. This implies that retry data should be included in the pretraining stage for practical LLM training, rather than in the finetuning (alignment) stage.

**Result 8: prepare fake retry data.** Retry data can be difficult to obtain, so we explore practical methods to automatically augment correct math solutions with "fake" mistakes, ideally without needing to semantically parse or understand the correct solution. The most effective method we found is to introduce a random future step $B$ as a "fake error" at each step $A$ in the solution, followed by $A$ as its "correction." This approach encourages the model not to skip steps, even though some solution steps may be interchangeable in order so these may not be true errors. In the synthetic iGSM setting, this method is nearly as effective as retry data with perfectly designed errors and corrections, and it could potentially be adapted to real-life math problems.

**Conclusion.** By utilizing fully-controllable synthetic data (e.g., controlling error rates or label masking), conducting controlled experiments (e.g., beam search vs. retry vs. error-free; pretrain vs. finetune), and performing fair comparisons (e.g., same number of training tokens), the goal of this paper is to try to predict the needs of future LLMs. We do not claim that the synthetic data used here can directly aid in building future LLMs. However, given that commercial LLMs already employ synthetic data (Team, 2024; Marah Abdin et al., 2024) and future LLMs are rumored to use $Q^\star$, it is perhaps crucial to understand how to best prepare and use such data effectively to teach models to learn from the mistakes. We give a more detailed conclusion section in Section 6.

---

[5]Beam search does not have the "error detection" subroutine but uses the model's next-token prediction probability distribution to simulate a re-generation process.

[6]To provide a strong comparison, for different $p$ we pretrain over the same number of tokens; so a higher $p$ means we pretrain over a smaller number of problems, because solutions with a larger $p$ are longer.

**(Problem)** The number of each Riverview High's Film Studio equals 5 times as much as the sum of each Film Studio's Backpack and each Dance Studio's School Daypack. The number of each Film Studio's School Daypack equals 12 more than the sum of each Film Studio's Messenger Backpack and each Central High's Film Studio. The number of each Central High's Film Studio equals the sum of each Dance Studio's School Daypack and each Film Studio's Messenger Backpack. The number of each Riverview High's Dance Studio equals the sum of each Film Studio's Backpack, each Film Studio's Messenger Backpack, each Film Studio's School Daypack and each Central High's Backpack. The number of each Dance Studio's School Daypack equals 17. The number of each Film Studio's Messenger Backpack equals 13. *How many Backpack does Central High have?*

**(Solution)** Define Dance Studio's School Daypack as p; so p = 17. Define Film Studio's Messenger Backpack as W; so W = 13. Define Central High's Film Studio as B; so B = p + W = 17 + 13 = 7. Define Film Studio's School Daypack as g; R = W + B = 13 + 7 = 20; so g = 12 + R = 12 + 20 = 9. Define Film Studio's Backpack as w; so w = g + W = 9 + 13 = 22. Define Central High's Backpack as c; so c = B * w = 7 * 22 = 16. *Answer: 16.*

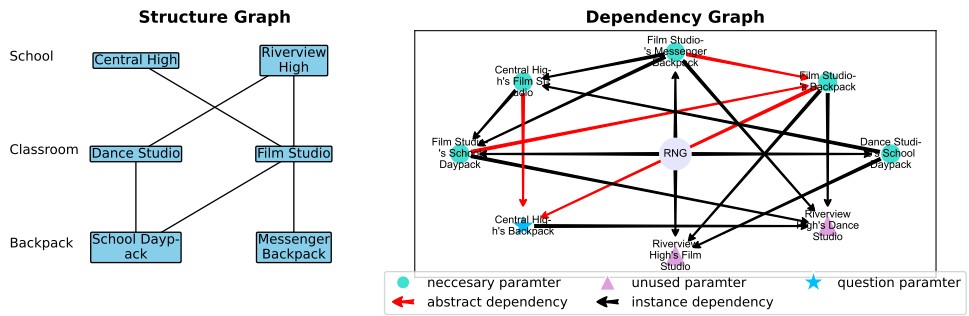

Figure 2: An example of a math problem with $\mathsf{op} = 7$ operations needed to compute its solution. A much harder example with $\mathsf{op} = 20$ is in Figure 9.

## 2 SYNTHETIC MATH DATA FROM PRIOR WORK

Ye et al. (2025) introduced a family of controllable, synthetic datasets of math problems with step-by-step solutions. These data simulate GSM8K (Cobbe et al., 2021), while removing arithmetic difficulties (by restricting computations to integers modulo 23) and common sense knowledge (e.g., a candle burns and its length decreases). What remains is "logic reasoning." The dataset has much larger diversity (over 90 trillion solution templates), and the solutions are fully verifiable. We briefly summarize it to make the paper self-contained, emphasizing some important aspects.

An example from their dataset is in Figure 2. The structure graph describes the set of *instance parameters*, such as "the number of school daypacks in each film studio." They also allow for *abstract parameters*, such as "the (total) number of backpacks in central high," which requires hierarchical computation.[7]

**The exact data construction is not important for this paper.** What matters is that the parameters form a *dependency graph*, as shown in Figure 2, where a parameter can be computed only when its predecessors have all been computed. To remove arithmetic difficulty, the computations are broken into binary operations — such as $12 + 13 + 7$ is broken into $(12 + 13) + 7$ — ensuring that failure to solve the problems is not due to arithmetic difficulties. They use $\mathsf{op}$ to denote the number of operations needed in the solution and prepared four families of data:

- iGSM-med$_{pq/qp}$ uses $\mathsf{op} \leq 15$ for train; $\mathsf{op} \in \{20, 21, 22, 23\}$ for OOD (out-of-distribution) test.
- iGSM-hard$_{pq/qp}$ uses $\mathsf{op} \leq 21$ for training and $\mathsf{op} \in \{28, 29, 30, 31, 32\}$ for OOD testing.

Here, $pq$ denotes the problem description comes before the question, and $qp$ otherwise. They also introduce reask data for evaluation purposes: for instance, iGSM-med$_{pq}^{\mathsf{op}=20,\mathsf{reask}}$ is a dataset constructed by first creating iGSM-med$_{pq}^{\mathsf{op}=20}$ and then re-sampling a parameter to query — this greatly changes their math data distribution, making it a good candidate for OOD evaluation. (After reask, the problem's $\mathsf{op}$ value may change.)

They showed that GPT-4/4o cannot solve such problems for $\mathsf{op} > 10$ (even with few-shot learning and their best efforts to remove English and arithmetic difficulties), indicating that the datasets are of some non-trivial difficulty. For completeness, we include in Figure 9 on Page 16 an example with $\mathsf{op} = 21$ to illustrate that these problems require non-trivial reasoning even for humans.

---

[7]In this example, it equals $A \times (B_1 + B_2)$ for $A$ = central high's number of film studios, $B_1, B_2$ = each film studio's number of school daypacks / messenger backpacks.

## 3 RESULT 0-1: LANGUAGE MODELS CAN RETRY UPON REGRET

Generative models solve math problems step by step in a chain-of-thought (CoT) manner. Each step in our math problems is a single sentence formatted as "Define [param] as X; so ...", as shown in Figure 2. How do generative models make mistakes in this CoT process?

The most *common* reasoning mistake[8] occurs when the model generates a [param] that is not yet ready for computation (i.e., the model has not determined the values of all parameters that [param] depends on, also known as "skipping steps", a frequent error even in GPT-4 (Bubeck et al., 2023)).

For example, in Figure 2, if the model generates "Define Central High's Film Studio" as the first few words in its solution, it cannot go back and erase this sentence, leading to a mistake. Ye et al. (2025) confirmed that not only do language models pretrained using the iGSM datasets make mistakes in this manner, but even GPT-4/GPT-4o make such mistakes (when using few-shot). It's worth noting that the failure example in Figure 1 is also in this spirit.

### 3.1 RESULT 0: MODELS CAN BE "REGRETFUL" AFTER MAKING MISTAKES

Interestingly, their same paper also implies the following:

> **Result 0.** *[corollary of (Ye et al., 2025)]For models pretrained on iGSM (with correct solutions only!), during their solution generation process, after writing "Define [param] as" for a wrong [param], they often* "realize" such a mistake, *showing a* regretful pattern *in their internal* states.

To see this, one can apply their probing technique (illustrated in Figure 3(a)) to extract information from the model's last hidden layer after "Define [param $A$] as" to see if the model knows $A$ can truly be computed next. This probing task is denoted as $\texttt{can\_next}(A) \in \{\text{true}, \text{false}\}$. They found:

- When $A$ ranges over all possible parameters, the probing 99% accurately predicts $\texttt{can\_next}(A)$, meaning the model knows if $A$ can be computed next, even for the hardest $\texttt{op} = 32$ problems.
- When the model makes a mistake, the first sentence with a mistake usually has $\texttt{can\_next}(A) = \text{false}$. Probing shows the model has $\sim$60% chance of knowing $\texttt{can\_next}(A) = \text{false}$, indicating it often knows it has made a mistake, *right after* stating the parameter name in full.[9]

This indicate that the model's internal states exhibit some "regretful" pattern, which can be detected via probing such as $\texttt{can\_next}$, which is almost just a linear classifier on top of its hidden states.[10] In other words, **error detection is easy** and is a skill almost already embedded within the model's internal states, even when pretrained on correct math problems only.

### 3.2 RESULT 1: LET MODELS RETRY UPON REGRET

If a model knows it is a mistake, why does it generate the wrong [param $A$] in the first place? The issue lies in the generation process. *Before explicitly stating* "Define [param $A$] as", the model might falsely think $A$ is ready to compute among *all the parameters* it can focus on. After stating it, the model shifts its focus to the actual computation of $A$, and this is the moment it can better realize that $A$ is not ready for computation (using its attention mechanism).[11] Now that we know the model exhibits some "regret" towards the mistake, can we use this to improve accuracy?

**Retry upon regret.** We conducted an experiment using the probing result to guide the model's generation process. After generating each solution sentence, we use the $\texttt{can\_next}$ probing to

---

[8]In practice, language models can make other mistakes such as in arithmetic or common sense (see Section 2); however, the design of the iGSM datasets has removed such difficulties, allowing us to focus solely on the reasoning aspect.

[9]Note that 60% accuracy is significant. If it were a random guess, 50% accuracy would be trivial. However, the probing method is 99% accurate in predicting true or false, and only on a small set of examples (i.e., when making mistakes), it has a 60% chance of correctly predicting $\texttt{can\_next}(A) = \text{false}$.

[10]This aligns with observations that detecting mistakes is usually easy: in works like (Liu et al., 2023; Li et al., 2023b; Wang et al., 2024), they show pretrained transformers can be easily fine-tuned to effectively verify their own solutions (true or false).

[11]Similar phenomenon also occurs in knowledge partial retrieval (Allen-Zhu & Li, 2025a).

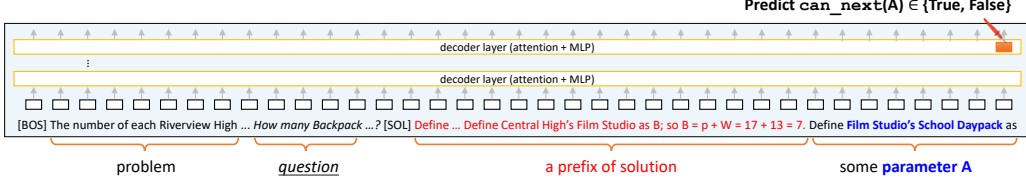

(a) `can_next` probing (Ye et al., 2025). After pretraining, V-probing technique can detect if the model's internal states exhibit a regretful pattern: right after "Define param $A$ as," the model no longer thinks $A$ is ready for compute.

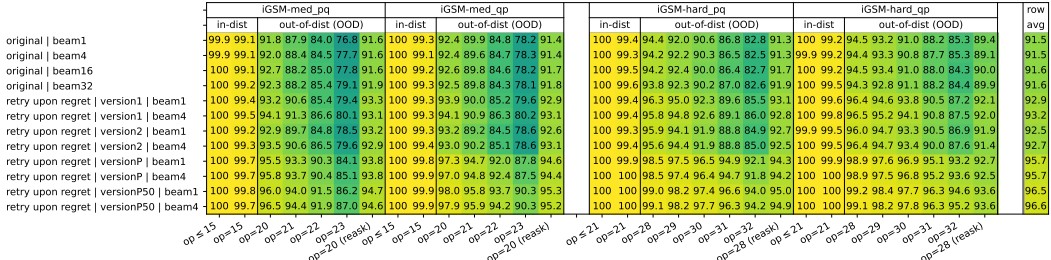

(b) Retry upon regret vs. original accuracies. Version1/2 uses the `can_next` probing to detect regret (see Section 3.2, they can detect errors to 99% accuracy), and versionP uses a perfect error detector to decide when to retry.

Figure 3: The `can_next` probing and using it to assist the model's generation process.

determine if the model knows it has made a mistake. If so, we revert to the end of the previous sentence and regenerate. We use multinomial sampling (i.e., beam=1 and dosample=true) during this regeneration process, with a maximum of 10 total retries for generating each solution.[12] We call this method "retry upon regret", and we carefully compare a few different versions (details are in Appendix G.2).

- The original `can_next` probing (Ye et al., 2025) freezes the pretrained model and fine-tunes a rank-8 update on the embedding layer — we call this "version1", and it can 99% accurately detect errors.[13]

- We additionally consider a different `can_next` probing that fine-tunes more parameters but with only 0.25% data — we call this "version2", and it can also 99% accurately detect errors.

- For comparison, we also consider the perfect, ground-truth `can_next` error detector and call this "versionP"; and we use "versionP50" to indicate using a maximum of 50 total retries.

The **goal** of *version1* is to check if we can use extremely small parameter changes from the pretrained model for error detection and use this towards higher accuracies. The **goal** of *version2* is to check if we can use a small number of fine-tuning samples for error detection. The **goal** of *versionP/P50* is for comparison purposes; after all, allowing perfect error detection can enable the model to correct nearly all mistakes if retried infinitely.

We compared this with the original generation (with beam=1, 4, 16, and 32); and we used beam=1 or 4 for all versions of "retry upon regret." The results give rise to the following:

> **Result 1** (Figure 3(b)). *The idea of "retry upon regret" can improve reasoning accuracy, to an extent better than beam search. However, the accuracy gain can be limited unless the error detector is highly accurate. This also alters the inference process and increases the inference complexity.*

We explain the three sentences in detail below:

- Comparing version1/2 with beam32, we see "retry upon regret" improves upon beam search. However, even though error detection is 99% accurate, this improvement is still marginal: about 2% for the $\mathsf{op} = 23$ (resp. $\mathsf{op} = 32$) case for iGSM-med (resp. iGSM-hard).

---

[12]It is necessary to limit the maximum number of retries to avoid infinite loops.

[13]See their reported probing accuracy table, where the $\texttt{can\_next}(A)$ probing accuracies can be 99% even for $\mathsf{op} = 23$ on iGSM-med or for $\mathsf{op} = 32$ on iGSM-hard.

> **(Solution - retry rate 0.5)** Define Dance Studio's School Daypack as p; so p = 17. ~~Define Film Studio's School Daypack as [BACK]~~. Define Film Studio's Messenger Backpack as W; so W = 13. ~~Define Central High's Classroom as [BACK]~~. ~~Define Central High's Backpack as [BACK]~~. Define Central High's Film Studio as B; so B = p + W = 17 + 13 = 7. Define Film Studio's School Daypack as g; R = W + B = 13 + 7 = 20; so g = 12 + R = 12 + 20 = 9. ~~Define Riverview High's Dance Studio as [BACK]~~. Define Film Studio's Backpack as w; so w = g + W = 9 + 13 = 22. ~~Define Riverview High's Dance Studio as [BACK]~~. Define Central High's Backpack as c; so c = B * w = 7 * 22 = 16.

(a) A solution example identical to Figure 2 but with retry_rate = 0.5. The strikethrough like "~~Define Central High's Backpack as~~" is for illustration purpose, and the actual data is normal English text without strikethrough symbols.

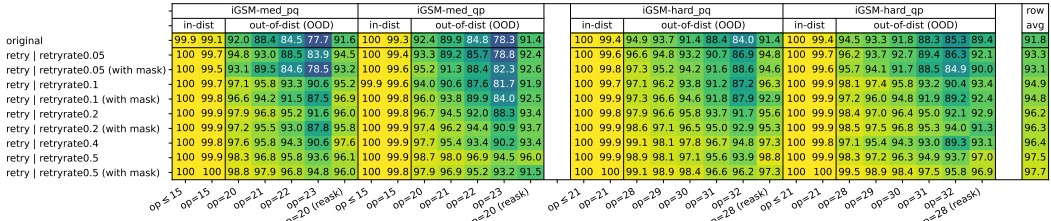

(b) Comparison of models pretrained using iGSM data with retry_rate > 0. For a stronger comparison, the model is pretrained on the retry vs original (error-free) data using the same number of tokens (i.e., retry data has fewer problems than original data) and identical training parameters, see Appendix G.1.

Figure 4: Pretrain language models on error-free vs retry data. **Observation:** especially on the hardest tasks (op = 23 or 32), models pretrained from retry data exhibit the greatest improvements for larger retry_rate.

- Comparing version1/2 with versionP, we see that the success of "retry upon regret" largely depends on an extremely accurate error detector — increasing the error detection success rate from 99% to 100% can significantly improve the final reasoning accuracy, but this is too ideal.[14]
- The idea of "retry upon regret" increases the inference complexity because one needs to keep an error detector model alongside and keep checking the correctness of the generated solution steps. In the event of an error, the model needs to regenerate using randomness (possibly multiple times) until it passes the error detector. Ideally, one wishes to have just one model to achieve "general intelligence" using the simplest autoregressive decoding algorithm, without multi-round error corrections.

# 4 RESULT 2-6: PRETRAIN WITH RETRY DATA

In this section, we prepare pretrain data to teach the model to directly correct mistakes.

**Math data with retry.** Since we use a controllable, synthetic math dataset, we can, at the beginning of each solution sentence, with probability retry_rate $\in [0, 1)$, insert a wrong parameter that cannot be computed next, followed by a special token [BACK].[15] We repeat this process, so with probability $(\text{retry\_rate})^2$, it may generate another wrong parameter at the same location, and so on. We provide an extreme example with retry_rate = 0.5 in Figure 4(a), and a more complex example with op = 21 and retry_rate = 0.2 in Figure 9. We call this the **retry data** (with error and correction), to distinguish that from the original (error-free) data.

- *Can a language model be pretrained using retry data to improve its reasoning accuracy compared with language models pretrained on the error-free data?*

We conduct experiments on the iGSM-med$_{pq/qp}$ and iGSM-hard$_{pq/qp}$ datasets for retry_rate ranging from 0.01 to 0.5. **For a controlled comparison**, we compare models when they are pretrained using the *same number of tokens* (along with other parameters).[16] This means, when pretrained with retry data, the model sees *fewer* math problems compared to the original training; when pretrained with larger retry_rate, the model also sees fewer math problems.

Another important question when pretraining with retry data is that:

---

[14]After all, a false negative in error detection results in a wrong answer, and having a false positive can result in the model regenerating too many times.

[15]This parameter is uniformly randomly chosen from all such parameters, except those already appearing.

[16]The same training parameters such as batch size, total steps, learning rates, see Appendix G. In particular, we adopt the learning rate from (Ye et al., 2025) (which was best tuned for the original error-free data) directly to the retry data.

- *Is it necessary to introduce label masks to prevent the model from learning the mistakes?*

Adding masking might make intuitive sense because we do not want the model to learn the mistakes — we only want it to learn to correct errors if needed. To answer this, we also perform controlled experiment to compare (1) standard auto-regressive pretraining and (2) pretraining with masking: adding label masks to ignore the auto-regressive loss on the wrong parameter tokens (i.e., ignoring text with ~~strikethrough~~ in Figure 4(a)).

> **Result 2-3.** *Our results in Figure 4(b) strongly support that:*
>
> - *Within a reasonable range,[17] **the more mistakes the better.** Especially on hard problems, such as on iGSM-med$_{qp}^{op=23}$, the accuracy jumps from 78% to 94% by using* retry_rate $= 0.5$.
>
> - ***Masking mistakes is unnecessary.** We observe that it is generally not needed to introduce label masking on the error data even for large* retry_rate $= 0.5$.

A reader seeing this for the first time may find the above results unbelievable: if the pretrain data is full of mistakes (even with retry_rate $= 0.5$), doesn't this interfere with the model's learning and encourage it to make lots of mistakes? To address this, we compute the statistics on how many times the pretrained model performs retries (i.e., how many [BACK] tokens it uses) during evaluation. We discover that (Figure 6 deferred to full paper):

> **Result 4** (Figure 6). *Models pretrained on retry data hardly retry (unless* retry_rate *is very high).*

For instance, Figure 6(a) shows if retry_rate $= 0.2$, even when pretrained without label masking, the model retries an average of $< 0.3$ times even for math problems with large op. This is because at each solution step, the retry data still has a higher $1 - 0.2 = 0.8$ chance to produce an error-free step. Thus, a model pretrained from such data is still incentivized to generate correct steps (e.g., using a low temperature).[18] If retry_rate $= 0.5$, this average retry count becomes $2 \sim 4$, but can be mitigated using label masking. Only on those wrong solutions does the model try its best to retry and fail in the end, see Figure 6(b).

Another concern with the retry data is whether the model can still output shortest solutions. Ye et al. (2025) discovered that language models can learn to produce shortest solutions without computing any unnecessary parameters.[19] Does the introduction of math data with errors increase the solution length? We discover that:

> **Result 5** (Figure 7). *Models pretrained on retry data can still output shortest math solutions.*

(Defer Figure 7 to full paper, which shows that when pretrained using retry data, model's generated solutions compute the same (small) number of unnecessary parameter as using original data.)

**Combining Results 2-5**, we see that it is safe to include retry data of this format as pretrain data to improve the model's reasoning accuracy. There is no change to the pretrain or inference process, no need to increase training time, and the solutions still satisfy many desirable properties.

Finally, it is tempting to compare retry data with beam search, which lets the model generate multiple solutions at each step and pick the most likely continuation based on the log-likelihood. As long as the model is pretrained on error-free data, even with 16 or 32 beams, its accuracy does not noticeable improve in all the cases (recall Figure 3(b)); while in contrast, if the model is pretrained with retry data, the accuracy easily goes up in Figure 4(b) (and this holds even for beam=1).[20]

---

[17]Naturally, retry_rate cannot approach 1. Exploring such extreme failure settings is not particularly interesting. For instance, retry_rate $= 0.5$ is already sufficiently extreme, indicating that half of the solution steps contain errors.

[18]Even when retry_rate $= 0.5$, if errors are sufficiently random (say, in a step there are 2 correct possibilities but 8 possible errors), the model is still incentivized to say correct sentences. This could be reminiscent of language model's learning on context-free grammars with mistakes (Allen-Zhu & Li, 2023): even though the model is trained *only* using data with grammar mistakes, it can output sentences correctly respecting the grammar at lower temperature.

[19]They discover the model achieves so via non-trivial mental planning to precompute the set of necessary parameters, before it starts to even generate the first solution sentence.

[20]To present the cleanest result, in Figure 4(b) we present the best accuracy among beam=1 or 4; one can still observe a high accuracy boost for beam=1 when pretrained with retry data.

Similarly, "retry upon regret" lets the model re-generate another solution step if error is detected, and this for instance gives only accuracy gain $78\% \Rightarrow 80\%$ in the $\mathsf{iGSM\text{-}med}_{pq}^{\mathsf{op}=23}$ case (recall Figure 3(b)), while pretrain with retry data can give $78\% \Rightarrow 95\%$. This is strong signal that:

> **Result 6.** *Error correction is a skill that can be **fundamentally different** from beam search or retry based on the model's randomness.*

Therefore, to truly improve the model's reasoning capability, it is crucial to modify the training data to include mistakes and their corrections.

## 5 RESULTS 7-8

Due to space limitation, we omit the technical details for Results 7-8 in this ICLR version to encourage readers to refer to our full paper at `ssrn.com/abstract=5250631`. We remark that the full paper underwent the ICLR 2025 review process, but we elected to present this camera-ready version as an *extended abstract*, aligning with the tradition in the theory community.

## 6 CONCLUSION

In this paper, we investigate whether language models can benefit from pretraining on data containing mistakes, followed by immediate error correction. Using a fully controllable synthetic setting, we demonstrate that models trained on such data outperform those trained on the same amount of error-free data.

In addition to the accuracy gain, Section 4 shows that using retry data is very safe: the model rarely makes mistakes even after pretraining with high error-rate retry data, and it is unnecessary to change the training process (simply autoregressive, no need to label-mask the errors). Retry data teaches models how to correct errors if needed, rather than encouraging mistakes.

It is important to note that such *error correction skill* does not come easily. A model pretrained with only error-free data cannot use (1) beam search or (2) retry based on error detection ("retry upon regret") to achieve comparable performance, see Section 3, unless the error detection is nearly perfect. This error correction skill is also very different from the original error-free reasoning and thus cannot be learned during parameter-efficient fine-tuning (PEFT) such as LoRA, see Result 7. This implies the necessity of adding retry data to the pretrain data for language models to truly learn the capability to correct errors.

While grade-school level math problems have many other difficulties (including arithmetic or common sense), following Ye et al. (2025), we have focused on the (logic-following) reasoning aspect, which is one of the weakest aspects of GPT-4.[21]

While it is unlikely that iGSM retry data will be directly used for pretraining future commercial-level LLMs, this paper aims to find guiding principles for necessary ingredients. We strongly discourage using fine-tuning to teach a model to correct errors (or using beam search, or letting the model regenerate upon encountering a mistake) as these are not effective. We advocate for adding mistakes and corrections at the pretrain level. While commercial LLMs use synthetic data (Team, 2024; Marah Abdin et al., 2024) and future LLMs are rumored to use $Q^\star$, it remains a question of how to prepare such synthetic data for the model to best learn error correction. Our Result 8 suggests that it is critical to teach a model not to skip steps. This can be done either through naively creating retry_weak data like ours or using more advanced prompting methods to encourage an auxiliary model to rewrite math data into such a format. We cannot explore such follow-up directions due to GPU resource limitations.

Finally, Part 2 of this work series focuses on how language models solve grade-school math level reasoning problems (including Part 2.1 (Ye et al., 2025)). We also cover how language models learn language structures in Part 1 (Allen-Zhu & Li, 2023) and learn world knowledge in Part 3 (Allen-Zhu & Li, 2024; 2025a;b), as well as how these may impact architecture design (Allen-Zhu, 2025).

---

[21] In contrast, arithmetic such as 10-digit multiplications can be (and perhaps should be) handled by calculators to save the model's capacity for other skills.

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
