# OpenReview forum: "Physics of Language Models: Part 2.2, How to Learn From Mistakes on Grade-School Math Problems"
_ICLR.cc/2025/Conference — ICLR 2025 Poster_

### Official Review · Reviewer_ZAFG · 2024-10-29

**Soundness:** 4
**Presentation:** 3
**Contribution:** 3
**Rating:** 6
**Confidence:** 4

**Summary:**

This work focus on understanding the usefulness of “error-correction” data into the pretraining stage. The experiment results show that error-correction data can improve the mathematical ability of the model more effectively than error-free data.

**Strengths:**

- The experiments are well-designed and rigorously conducted.
- The finding that “even when model is pretrained on retry data with high error rate, it does not tend to produce erroneous steps” is interesting.

**Weaknesses:**

- Some previous work[1] has already pointed out that error-correction data enables LLM to achieve higher reasoning accuracy compared to error-free data.
- This work only evaluates on IGSM dataset, which is a synthetic data. It would be more convincing to also experiment on realistic mathematical datasets.

[1] Learning from mistakes makes llm better reasoner

**Questions:**

- Will the experiment code and data be open sourced?
- In section 3.1, as the can_next is “a linear classifier on top of its hidden states”, is the one who “knowing can_next(A)=false” actually the linear classifier rather than the model itself? Does the author mean that because the hidden states of the model show a distribution that can be accurately predicted by the classifier, the model "knowing can_next(A)=false"?
- It would be appreciated if the authors provide some insights about why “the slightly more complex retry miss data does not improve accuracy by much”

---

> ### Author Response · Authors · 2024-11-28
> **Response to Reviewer ZAFG**
>
> We sincerely thank the reviewer for liking our experiment design, and for the time carefully reading this paper. To address your questions:
>
> **Question**:
>
> > Will the experiment code and data be open sourced?
>
> **Answer**:  We are in the last stage of releasing the code. We’ve done code refactoring, but are bound by legal issues so are waiting for the approval (from all of our authors’ affiliation).
>
> **Question**:
>
> > In section 3.1, as the can_next is “a linear classifier on top of its hidden states”, is the one who “knowing can_next(A)=false” actually the linear classifier rather than the model itself? Does the author mean that because the hidden states of the model show a distribution that can be accurately predicted by the classifier, the model "knowing can_next(A)=false"?
>
> **Answer**:  Not so to your first question! The hidden states computed by the model are still important. And, as you suggest in the second question, the hidden states have “shown a distribution that can be accurately predicted by the classifier”. In other words, we are saying that after pretraining, the transformer model’s architecture can not only generate next tokens, but also its internal states have stored some specific pattern that can be directly used for error detection.
>
> (If you look at our Anonymous prior work, attached as supplementary result, see its Figure 6a, if one replaces such pretrained transformer with a randomly-initialized transformer, then such (almost) linear probing will not give you good can_next accuracies. This means error-detection is not as trivial as a linear classifier on its own - even if a neural tangent kernel is involved.)
>
> **Question**:
>
> > It would be appreciated if the authors provide some insights about why “the slightly more complex retry miss data does not improve accuracy by much”
>
> **Answer**:  There are typically two types of mistakes in such iGSM data, (1) one is to compute a step not ready for computation (the focus of this paper), and (2) the other is to compute some unnecessary parameter, not needed towards the final answer. This later “mistake” is more like a WARNING but not ERROR, because computing redundant quantities doesn’t impact the final accuracy. (See our supplementary material, last page, for experiments of GPT-4o that make similar two types of mistakes).
>
> Our belief is that, when using retry_weak, the model has learned to not to skip steps before computing any necessary quantity. In other words, the data focuses on teaching the model not to make mistakes (1). In contrast, retry_miss has tried to be “smart” in terms of parsing, but actually has introduced many unnecessary parameters into the candidate solutions, thus it teaches the model not to do (1) or (2) simultaneously.
>
> Now, because (1) is directly linked to the accuracy, and also because (2) happens more rarely (much rarer comparing to mistake (1), see our prior work in supp materials), focusing only on (1) is better.
>
> If you want any experimental support, please check our Figure 11b on page 17, in Line 897-899, you can see that retry_miss yields solutions that have slightly fewer redundant computations, compared to retry_weak.
>
>
> **Question**:
>
> > Some previous work[1] has already pointed out that error-correction data enables LLM to achieve higher reasoning accuracy compared to error-free data. This work only evaluates on IGSM dataset, which is a synthetic data. It would be more convincing to also experiment on realistic mathematical datasets.
>
> **Answer**:  Sure and we can add a citation to [1]. Indeed there are many trials to pretrain models on error-correction, but in most cases such error-correction data comes from GPT-4 or similar models, while we also know that GPT-4o is bad at correcting its own mistake (perhaps less so for o1). We are in a different focus here, that is we wish to study the difference between error detection vs correction, and the necessary training steps needed if such data is present.
>
> For instance, the recent Reflection model (https://huggingface.co/mattshumer/Reflection-Llama-3.1-70B) is a well-known attempt to do error correction, but most people believe they have failed. What they do is to finetune an existing model on error-correction data, and our conjecture is that this may not work — precisely for the reason we discovered in this paper.
>
> In other words, we wish to use the synthetic data to perform more controlled experiment, to perhaps make some predictions. Not only about what type of training is required, but also perhaps what format of the data is encouraged. Even though we cannot reliably produce large number of such error-correction data using GPT-4o, but perhaps tomorrow with GPT-o1 or other models, we shall be able to do so.
>
> Thanks again for reading this paper!

---

### Official Review · Reviewer_SYew · 2024-11-01

**Soundness:** 3
**Presentation:** 2
**Contribution:** 2
**Rating:** 5
**Confidence:** 3

**Summary:**

The paper aimed at improving language model accuracy in reasoning tasks (grade-school math) by pretraining with "error-correction" where errors are followed by immediate corrections, to teach models self-correction as they generate outputs. The authors show that pretraining with error-correction data boosts reasoning accuracy, without multiple rounds of prompting.

**Strengths:**

- The paper proposed an interesting idea by incorporating error-correction data directly in pretraining, rather than relying on post-generation prompting for correction with empirical evidence of improved performances.
- The authors compared different experimental settings, such as “retry upon regret,” masking, and pretraining versus fine-tuning with retry data.

**Weaknesses:**

- The paper has unclear justification for synthetic data choice.
- Writing and the structure of the paper is very unclear and hard to follow. For example, the experimental setting or conclusion section is not contained in the paper.
- While the paper presents extensive experimental results, the paper primarily focuses on using one model, it is unknown if such techniques are generalized to different models (e.g., LLAMA, Gemma, Mixtral etc.). The paper further lacks comparisons to existing methods.

**Questions:**

The paper would benefit significantly from a reorganization into a structure that aligns more closely with the familiar flow of this venue. Specifically, it would help to present the experimental settings early in the paper, followed by a dedicated focus on results and analysis. Additionally, the authors could simplify the tables to better highlight key findings, and consider using plots to present some of the results for improved clarity and reader comprehension.

---

> ### Author Response · Authors · 2024-11-28
> **Response to Reviewer SYew**
>
> We sincerely thank the reviewer for the time reading this paper, and we are sorry to see that our writing did not meet your expectation. For instance, due to the large number of results we wish to present, we have shortened the conclusion to only 8 lines, included in Line 168-175 of this paper. We can move this to page 10 if you suggest us to.  We can also move the experimental figures to early as you requested.
>
> > Additionally, the authors could simplify the tables to better highlight key findings, and consider using plots to present some of the results for improved clarity and reader comprehension.
>
> Can we kindly ask the reviewer to give some concrete suggestions on which table can better be converted into plots? The issue is that the columns are for different tasks so they cannot be connected by a curve; the rows are for different data settings so may not be connected by a curve either. We have tried to color-code all the tables to make it clear which model/data works the best, but would always love to hear other suggestions.
>
> **Question**:
>
> > it is unknown if such techniques are generalized to different models (e.g., LLAMA, Gemma, Mixtral etc.). The paper further lacks comparisons to existing methods.
>
> **Answer**:  Many recent controlled studies have shown that decoder-only model architectures equipped with full attention and rotary embedding behave very similarly to each other (2305.13673, 2309.14402, 2404.05405). They have minor differences, but not much to fundamentally change the model’s learnability behavior. With this said, we agree that it can still be beneficial to repeat the experiments with other architectures. In Line 917 of this submission in the appendix, we mentioned that we also played with the Llama architecture (whose main difference to GPT2 is the gated MLP), and did not see any major difference. We didn’t include a table in this paper unfortunately, because we are bound by GPU resources (+ a sudden job loss), so cannot afford re-doing all the experiments in this paper with Llama. If this paper luckily gets in, we hope to include a complete set of Llama experiments before camera ready. Thanks.

---

### Official Review · Reviewer_7os6 · 2024-11-04

**Soundness:** 4
**Presentation:** 3
**Contribution:** 3
**Rating:** 8
**Confidence:** 3

**Summary:**

This paper studies the impact of training language models with "error-then-retry" format data on the reasoning performance. With experiments on a synthetic GSM8k-style math reasoning datasets, the authors conclude that pretraining with such data improves reasoning accuracy than on same amount of "error-free" normal data, and LoRA fine-tuning with such data does not help.

**Strengths:**

1. I like how the authors approach this problem in a data-centric, rigorous way: controlled experiments on synthetic data.
2. Experiments are solid

**Weaknesses:**

The paper mainly experiments with one type of error: inserting a wrong parameter that cannot be computed next. While this is easy to implement, it would be hard to simulate all kinds of errors that language models can make in real-world reasonings scenarios. This cast doubts on how the suggested approach can be deployed to train LMs on non-synthetic data. If the authors could provide some discussion on how the method can be generalized to different errors (e.g., math calculation error, context misunderstanding, ...) in a scalable and controllable way, that would be very promising.

**Questions:**

I would like to hear the authors' thoughts on the inference-time scaling properties of model trained with "retry" data: given a fixed inference-time computation budget, is it better for model to produce reasoning chains with retry (more tokens to reach final answer, but average accuracy is higher) or without retry (less tokens to reach final answer, but average accuracy is lower)?

---

> ### Author Response · Authors · 2024-11-28
> **Response to Reviewer 7os6**
>
> We sincerely thank the reviewer for the support of this work and the time carefully reading this paper.
>
> To address your question, our thoughts are two-fold. First, we believe this may not have a clear answer unless the model architecture is fixed. Full attention gives the best possible in-context reasoning capability because it sees everything in the past; while Mamba/SSM/S4/LSTM/… can afford longer context length but may not be able to reason well when there are too many inference-time tokens. We think it may be very beneficial to study what’s the right tradeoff here, and perhaps even what’s a best architecture that can support long reasoning chains to the greatest extent. This may require some totally different study and is beyond the focus of this paper.
>
> Second, say that the context length is not an issue (e.g., reasoning chain within 2000 tokens and full attention is used). Then, we believe two things would matter. One is the pretrain/finetune data - if you don’t have error correction or detection data then you probably can’t do anything good. If you have error detection data, then you can train an error-detector and do things like “try upon regret”, or even more advanced ones such as generation using beam=100 and then consensus vote using error detector. Only when you have error correction data, can you start to compare the two choices like you said.
>
> But even in such a case, the two choices may not be mutually exclusive. If you train with retry data, you can also generate multiple times (say beam=100) and do consensus vote. We view this as sort of orthogonal to the inference-time computation budget. If the budget is high, and if you do have retry data, then we recommend doing both.
>
> Thanks again for raising this question!

---

> > ### Comment · Reviewer_7os6 · 2024-11-30
> > **Response**
> >
> > Thank you for your response!

---

> > > ### Author Response · Authors · 2024-12-03
> > >
> > > Thanks for your time! Very much appreciate it.

---

### Official Review · Reviewer_ZAr2 · 2024-11-04

**Soundness:** 3
**Presentation:** 4
**Contribution:** 3
**Rating:** 8
**Confidence:** 4

**Summary:**

This paper improving math reasoning in LLMs by adding error-correction data directly into pretraining, instead of using the usual multi-round prompting---that dominant self-refine approach. Using a synthetic math dataset, the authors show that training with examples of mistakes followed by corrections leads to better reasoning accuracy, even beating models trained on error-free data. The study tries to answer some interesting question: how to prepare error-correction data, is finetuning sufficient to learn self-correction or is pretraining necessary and whether how this approach compares to beam search. Although on a synthetic and a very controlled setup, the results present some fresh perspective into pretraining LLMs to do better revisions.

**Strengths:**

- The paper is very well written. Although the results are dense, the authors did a good job summarizing and condensing the main takeaway messages.

- The studied problem is interesting: We still don’t fully understand how to pretrain LLMs for effective reasoning from the ground up. This paper explores self-correction within the pretraining phase, a fresh perspective that hasn’t been widely explored in the literature, aside from a few works like Quiet-STaR.

- The experiments and ablations are well-designed. The authors clearly state their research questions early on and address them in a logical sequence.

**Weaknesses:**

My main concern with this paper is the uncertainty around whether these findings will generalize to practical LLM pretraining scenarios. I'll expand on some specific limitations below:

- The types of problems in the i-GSM dataset don’t reflect real-world reasoning tasks. Specifically, every reasoning step required here is limited to a single type of computation—finding the next node in a directed acyclic graph (DAG) and calculating its value. But what about other reasoning types where models need to compare values, handle ambiguity, or apply commonsense knowledge? Although I appreciate the focus on math reasoning, can the authors confidently assert that these results will apply to more complex, realistic reasoning tasks?

- Current LLMs struggle with error detection, as the authors note in the introduction. However, their findings in L259-260 suggest that error detection can be effortlessly embedded within the model’s internal states. This may be due to the task’s synthetic nature, where the model could have learned to encode specific errors, like “skip step,” in its parameters. But this is unlikely to generalize to other errors. For example, could the model's hidden states reliably detect other error types, like incorrect calculations?

- The paper’s fine-tuning experiment is limited to simple LoRA tuning. What about full fine-tuning using a fraction of the pretraining data? The authors mention (L484-485) that the cost of full fine-tuning would match pretraining with retry data, but this wouldn't hold if we fine-tune with just a fraction of the pretraining data. Would the results remain consistent in that scenario?

- I’d expect more discussion on inference-time algorithms and their impact on performance. If I’m following correctly, most experiments use greedy decoding or occasionally beam search. It would be insightful to understand how additional inference-time resources—like generating more tokens or applying consensus voting—might affect error correction. Result 4 (concerning model retries) is based on greedy decoding; how would this result change with sampling?

**Questions:**

See weaknesses above

---

> ### Author Response · Authors · 2024-11-28
> **Response to Reviewer ZAr2 [1/2]**
>
> We first would like to thank the reviewer for liking the studied problem and our presentation. Let us try to address your questions below.
>
> **Question**:
>
> > The types of problems in the i-GSM dataset don’t reflect real-world reasoning tasks. … But what about other reasoning types where models need to compare values, handle ambiguity, or apply commonsense knowledge? … can the authors confidently assert that these results will apply to more complex, realistic reasoning tasks?
>
> **Answer**:  This is a very complex question so let us break it down.
>
> First, we believe there are many LLM issues that can be broadly thought of as general “reasoning”.
> * This includes commonsense knowledge, where one retrieves some factual knowledge and does calculations on top of that. Prior work (arxiv 2309.14402) showed that LLMs can struggle even on single-steps of such calculations.
> * This includes arithmetic operations (such as comparing values), and there is rich literature that tries to tackle the arithmetic capability of LLMs.
> * This includes ambiguity, where this paper has tried (but not fully) touched that, by creating data that has both “A’s backpacks” and “A’s messenger backpacks” to try to create some ambiguity.
>
> We believe it can be beneficial to study LLMs on such issues separately. For commonsense knowledge, LLMs struggle for a totally different reason (see 2309.14402); for arithmetics, the token embedding (i.e., order of digits, starting from higher vs. lower digit) can significantly impact models' behavior (see 2402.03822 and many earlier works as well). We feel that studying DAG is one of the first steps towards this direction, and for instance, this may connect well with the (current failure of) Reflection models (see https://huggingface.co/mattshumer/Reflection-Llama-3.1-70B), where the authors proposed to use <reflection> (similar to our [BACK] token) to fine-tune a model for auto error-correction. To the best of our knowledge, they may have failed, and we conjecture the reason is precisely in this paper - you need to add such data to the pretrain stage.
>
> **Question**:
>
> > Current LLMs struggle with error detection … L259-260 suggest that error detection can be effortlessly embedded within the model’s internal states. This may be due to the task’s synthetic nature … But this is unlikely to generalize to other errors. For example, could the model's hidden states reliably detect other error types, like incorrect calculations?
>
> **Answer**:  We also break this down to a few aspects.
>
> First, the failure of internet-data pretrained LLMs on error detection could be due to many reasons. A most probable reason could be because of lack of such training data (to cover all types of reasoning errors). In our specific iGSM data setting, although we claimed that the “regret” is embedded in the model weights, it also requires labelled data in order to find out “how” such regret is embedded. Our submitted paper mainly distinguishes the difficulty of error detection vs. correction (even when abundant such labeled data is present), and we agree that error detection can also be hard on its own (when there’re very few training data).
>
> Second, this could **generalize to other errors**. In the arithmetic calculation case as the reviewer pointed out, when adding two numbers like 452352 and 547647, if the model uses single-digit tokenization, it could potentially first output 1 and then followed by six digits and regret for doing so. By the way, the correct answer is 999999 and Llama3.1-8B and 70B both output 1000000 (we just tried). Also, humans when solving math problems sometimes try one method and then only later (after some derivations) realize that the method doesn’t work.
>
> Third, the fact that the internal state showed "regretful" pattern should not be too surprising. We are not saying the model (automatically) knows how to correct such mistakes, but saying that the model may have known that a mistake is made (via probing). In our DAG case, after a parameter name is written, the model needs to continue on the math calculations and at this point it may realize that some parameter it depends on is not ready. This behavior can happen in many real-life error scenarios as well as we mentioned above.

---

> > ### Comment · Reviewer_ZAr2 · 2024-12-01
> >
> > Thank you for the detailed response and additional experiments. I will maintain my positive score.

---

> > > ### Comment · Reviewer_ZAr2 · 2024-12-02
> > >
> > > After some thought, I think the conference could definitely benefit from the paper experiments and rigor as long as the authors admit the synthetic nature of their experiments, which they do. Therefore, I have decided to raise my score to 8.

---

> > > > ### Author Response · Authors · 2024-12-03
> > > >
> > > > Thank you so much for your support. We truly hope this paper can get it so we can better focus on the follow-ups!

---

> ### Author Response · Authors · 2024-11-28
> **Response to Reviewer ZAr2 [2/2]**
>
> **Question**:
>
> > What about full fine-tuning using a fraction of the pretraining data? The authors mention (L484-485) that the cost of full fine-tuning would match pretraining with retry data, but this wouldn't hold if we fine-tune with just a fraction of the pretraining data. Would the results remain consistent in that scenario?
>
> **Answer**:  That’s a great question. We initially tried this, but did not include it due to its poor performance. Now since the reviewer has asked, we are adding a set of such experiments and reporting the result. To be clear, here, we are not talking about **early stopping** and early stopping doesn't work either. In fact, all of our experiments in this submission are already reporting the best-checkpoint accuracy (not the last-checkpoint accuracy).
>
> The correct way to do this is to reduce the full finetuning time from T to T'<<T and use its own learning-rate decay scheduling with respect to T'.  In our **updated supplementary materials**, we have included newFig10.pdf (an updated Figure 10), which includes the results for choosing T' = T/10 and for retryrate = 0.05, 0.1, 0.2, 0.5.
>
> **In short, the summary is** that "pretrain with no error for time T, then full finetuning with T/10 of the retry data" (a.k.a. continued pretrain x 0.1) works on par with LoRA finetuning, no better (often worse) than directly pretraining with T of retry data --- and a lot worse than T+T (continued pretrain). Thus, **our main claim remains very valid**: unlike error detection, error correction requires a lot of weight change and that's why LoRA fails and also why full finetuning needs a large number of steps **(even T/10 is not enough)**. In contrast, for error detection it is very cheap (both in terms of weight updates and the finetune steps **can be as low as T/400**, see line 305-306).
>
> **Question**:
>
> > I’d expect more discussion on inference-time algorithms and their impact on performance. If I’m following correctly, most experiments use greedy decoding or occasionally beam search. It would be insightful to understand how additional inference-time resources—like generating more tokens or applying consensus voting—might affect error correction. Result 4 (concerning model retries) is based on greedy decoding; how would this result change with sampling?
>
> **Answer**:  As we discussed in the paper, generating more tokens and then do sampling (namely, beam search with dosample=True) doesn’t work. As for consensus voting, this (more or less) needs the model to have the capability to detect errors and thus needs additional finetune samples (see our response to your first question). In some sense, “retry upon regret” is our very baby implementation of this process, that is to use an error-detector to do consensus voting (with only one candidate). We believe adding more candidates would help, but it heavily depends on the error-detection correctness and may deviate us from the main focus of this paper.
>
> As for your mention of Result 4, retries will start to occur more if you do sample with *high* temperatures. This is expected, and as we noted in Footnote 15, actually this same phenomenon was studied carefully in prior work 2305.13673 for the context of grammar mistakes. In short, if the pretrain data distribution teaches the model to make error with say 50% of chance, then when temp=1, the model is actually learning to do this also at 50% error rate. We feel that it is not needed to repeat this experiment (i.e., when temp varies) as much of the work was done in prior work 2305.13673.
>
> We hope our detailed reply has resolved some of your concerns. Thanks again!

---

### Meta-Review · Area_Chair_ScPm · 2024-12-21

**Metareview:**

(a) Scientific Claims and Findings
The paper claims that training language models with error-correction data during pretraining can significantly improve their reasoning capabilities. The study evaluates the impact of such pretraining on a synthetic math dataset and compares it against error-free data. It also investigates the role of error-detection and correction, asserting that pretraining is more effective than fine-tuning for error-correction tasks.

(b) Strengths
- Novel focus on integrating error-correction directly into the pretraining phase.
- Rigorous experimental design, including well-defined ablations and investigations of key factors like masking and error rates.
- Results are clearly reported, showing meaningful improvements over baseline models trained on error-free data.

(c) Weaknesses
- The experiments are primarily conducted on a single model, raising concerns about the generalizability of the proposed techniques to other model architectures (e.g., LLAMA, Gemma, Mixtral).
- Heavy reliance on synthetic datasets limits generalizability to real-world reasoning tasks.
- Limited discussion of scalability and applicability to broader error types in language models.
- Sparse comparisons with alternative approaches, including inference-time techniques.

(d) Decision
While the synthetic nature of the experiments limits generalizability, the paper provides a fresh perspective on error-correction pretraining and its potential to enhance reasoning accuracy. The positive reception from reviewers and the clear presentation of insights justify an acceptance.

**Additional Comments On Reviewer Discussion:**

During the rebuttal, reviewers raised concerns about the synthetic nature of the dataset and the generalizability of the findings to real-world tasks. The authors addressed these by acknowledging the limitations and emphasizing the exploratory nature of the study. Additional experiments (e.g., fine-tuning with a fraction of pretraining data) were provided, which reinforced the authors' claims that error correction requires extensive pretraining rather than fine-tuning.

Reviewers also questioned scalability to other error types and inference-time tradeoffs. The authors clarified that while this paper focuses on specific synthetic tasks, their approach can be extended to real-world scenarios with adequate data. Discussions about inference-time algorithms highlighted the orthogonality of their method to other decoding strategies.

Reviewer sentiment improved during the rebuttal, with one reviewer increasing their score after being convinced by the authors' arguments.

As for the weakness of generalizability across architectures, the authors argued that many recent studies have demonstrated that decoder-only model architectures equipped with full attention and rotary embeddings behave similarly in terms of learnability. While minor architectural differences exist, these are unlikely to fundamentally alter the proposed method's effectiveness. They mentioned having tested their approach on the LLAMA architecture (which includes gated MLP) and observed no major differences in results compared to their primary model. However, they did not include detailed results in the paper due to resource constraints.

---

### Decision · Program_Chairs · 2025-01-22

Accept (Poster)